# Multifaceted Assessment of Wastewater-Based Epidemiology for SARS-CoV-2 in Selected Urban Communities in Davao City, Philippines: A Pilot Study

**DOI:** 10.3390/ijerph19148789

**Published:** 2022-07-19

**Authors:** Maria Catherine B. Otero, Lyre Anni E. Murao, Mary Antoinette G. Limen, Daniel Rev A. Caalim, Paul Lorenzo A. Gaite, Michael G. Bacus, Joan T. Acaso, Refeim M. Miguel, Kahlil Corazo, Ineke E. Knot, Homer Sajonia, Francis L. de los Reyes, Caroline Marie B. Jaraula, Emmanuel S. Baja, Dann Marie N. Del Mundo

**Affiliations:** 1Department of Clinical Epidemiology, College of Medicine, University of the Philippines Manila, Ermita, Manila 1000, Philippines; mbotero@up.edu.ph (M.C.B.O.); esbaja@up.edu.ph (E.S.B.); 2College of Medicine Research Center, Davao Medical School Foundation, Inc., Bajada, Davao City 8000, Philippines; 3Department of Biological Sciences and Environmental Studies, University of the Philippines Mindanao, Mintal, Davao City 8000, Philippines; lemurao@up.edu.ph (L.A.E.M.); dacaalim@up.edu.ph (D.R.A.C.); jtacaso@up.edu.ph (J.T.A.); rmmiguel@up.edu.ph (R.M.M.); 4Philippine Genome Center Mindanao, University of the Philippines Mindanao, Mintal, Davao City 8000, Philippines; pagaite1@up.edu.ph (P.L.A.G.); mgbacus@up.edu.ph (M.G.B.); 5Marine Science Institute, University of the Philippines Diliman, Diliman, Quezon City 1101, Philippines; malimen@msi.upd.edu.ph (M.A.G.L.); cjaraula@msi.upd.edu.ph (C.M.B.J.); 6Project Accessible Genomics; kngcorazo@addu.edu.ph (K.C.); ineke@inekeknot.nl (I.E.K.); homer.sajonia.ii@gmail.com (H.S.II); 7Biology Department, Ateneo de Davao University, Roxas Avenue, Davao City 8000, Philippines; 8Institute for Biodiversity and Ecosystem Dynamics, University of Amsterdam, 1012 WX Amsterdam, The Netherlands; 9Department of Civil, Construction, and Environmental Engineering, North Carolina State University, Raleigh, NC 27207, USA; fldelosr@ncsu.edu; 10Institute of Clinical Epidemiology, National Institutes of Health, University of the Philippines Manila, Ermita, Manila 1000, Philippines; 11Department of Food Science and Chemistry, University of the Philippines Mindanao, Mintal, Davao City 8000, Philippines

**Keywords:** COVID-19, Philippines, public health surveillance, SARS-CoV-2, wastewater-based epidemiology, whole genome sequencing

## Abstract

Over 60 countries have integrated wastewater-based epidemiology (WBE) in their COVID-19 surveillance programs, focusing on wastewater treatment plants (WWTP). In this paper, we piloted the assessment of SARS-CoV-2 WBE as a complementary public health surveillance method in susceptible communities in a highly urbanized city without WWTP in the Philippines by exploring the extraction and detection methods, evaluating the contribution of physico-chemical–anthropogenic factors, and attempting whole-genome sequencing (WGS). Weekly wastewater samples were collected from sewer pipes or creeks in six communities with moderate-to-high risk of COVID-19 transmission, as categorized by the City Government of Davao from November to December 2020. Physico-chemical properties of the wastewater and anthropogenic conditions of the sites were noted. Samples were concentrated using a PEG-NaCl precipitation method and analyzed by RT-PCR to detect the SARS-CoV-2 N, RdRP, and E genes. A subset of nine samples were subjected to WGS using the Minion sequencing platform. SARS-CoV-2 RNA was detected in twenty-two samples (91.7%) regardless of the presence of new cases. Cycle threshold values correlated with RNA concentration and attack rate. The lack of a sewershed map in the sampled areas highlights the need to integrate this in the WBE planning. A combined analysis of wastewater physico-chemical parameters such as flow rate, surface water temperature, salinity, dissolved oxygen, and total dissolved solids provided insights on the ideal sampling location, time, and method for WBE, and their impact on RNA recovery. The contribution of fecal matter in the wastewater may also be assessed through the coliform count and in the context of anthropogenic conditions in the area. Finally, our attempt on WGS detected single-nucleotide polymorphisms (SNPs) in wastewater which included clinically reported and newly identified mutations in the Philippines. This exploratory report provides a contextualized framework for applying WBE surveillance in low-sanitation areas.

## 1. Introduction

Following the World Health Organization’s declaration of a global pandemic in March 2020, public health surveillance systems at the national and local levels were immediately instituted to reduce the transmission of COVID-19. Surveillance aims to rapidly detect, isolate, and manage cases and their contacts, guide control measures, and monitor the epidemiologic trends and evolution of SARS-CoV-2 [1]. In the Philippines, COVID-19 surveillance relies heavily on RT-PCR-based clinical diagnostics in symptomatic individuals and their contacts. However, clinical monitoring and contact tracing are limited in the early detection or prediction of community outbreaks and can be logistically demanding when applied to a large population [2,3]. Therefore, when interventions aim to curb transmission from patients with no symptoms or mild infection, complementary public health surveillance methods that can provide effective and faster results of community-level infection data using fewer resources must be explored.

SARS-CoV-2 is shed in the feces of symptomatic and asymptomatic COVID-19 cases [4,5,6]. Over 50 countries utilized sewage wastewater monitoring for COVID-19, whether in national, regional, or localized sites [7]. Studies from the Netherlands and the USA showed that wastewater-based epidemiology (WBE), or wastewater-based surveillance, can detect and predict hotspots of COVID-19 infections earlier than clinical surveillance [8,9]. Larsen and Wigginton [10] further projected that WBE could give a seven-day lead over COVID-19 case reporting in symptomatic or clinical surveillance. Thus, WBE can serve as an early-warning public health tool that can help control the spread of COVID-19, especially in areas where diagnostic resources are limited [11]. WBE can also be utilized in assessing control strategies against COVID-19 at the community level. SARS-CoV-2 viral titers in WBE declined in response to social isolation and lockdowns in Montana, USA [12], Paris, France [8], and Rio de Janeiro, Brazil [13]. In Brazil [13] and Singapore [14], WBE data from samples from sewer pipes were used to determine where public health interventions should be intensified. Circulating variants and strains of SARS-CoV-2 were identified in wastewater using whole-genome sequencing (WGS), supporting wastewater surveillance for tracing variant importations and circulation [15,16,17].

Davao City is the third-most populous Highly Urbanized City (HUC) in the Philippines, with 1.8 million inhabitants. It is considered as the gateway for trade in the Southern Philippines [18]. From 8808 confirmed cases of COVID-19 in Davao City at the end of 2020, as of 16 January 2022, there are 56,398 cases coming from the 182 *barangays* (or the smallest political unit in the Philippines) [19]. Like many cities in the Philippines, Davao City does not have a centralized wastewater management system. As a result, untreated and inadequately treated domestic wastewater and sewage from private and communal septic tanks may be released into rivers and creeks along with surface run-off [20,21]. COVID-19 WBE in low-sanitation settings in Ecuador [20] and portions of Brazil [13] demonstrated how WBE data could be used to design public health interventions and monitor COVID-19 transmission. In addition, wastewater particle composition may vary among wastewater sampling locations, and their associations with viruses in wastewater may affect viral decay [22]. Moreover, the individual and combined effects of physico-chemical, hydrologic, and anthropogenic factors on viral decay in the wastewater matrix must be explored.

This study piloted and assessed WBE and WGS as methodologies for COVID-19 surveillance in selected urban communities in Davao City, Philippines. We also provided insights on the possible contributions of physico-chemical, hydrologic, and anthropogenic factors of wastewater on the RNA preservation. We also identified the logistical and environmental challenges in conducting wastewater-based surveillance in low-resource and low-sanitation communities.

## 2. Materials and Methods

### 2.1. Study Sites

This descriptive pilot study was implemented in Davao City, a chartered city in the Davao Region, the southern portion of the Philippines (Figure 1). The City Government of Davao releases weekly COVID-19 transmission risk categories (CRITICAL, HIGH, MODERATE, or LOW) for its 182 *barangays*. These weekly characterizations are based on new and active cases normalized against the Weekly or Average Daily Attack Rate (ADAR) and the two-week Growth Rate (2WGR) for COVID-19 [23]. Guided by this weekly COVID-19 Risk Assessment *Barangay* Classification and the expert advice of the City Health Office, six communities representing HIGH to MODERATE risk of COVID-19 transmission were identified as sample collection sites (Figure 1). One wastewater sampling site within each *barangay* was determined with the help of the *barangay* official or their designated personnel. Accessibility and presence of a continuous flow of wastewater were considered in choosing the site. 

### 2.2. Review of Secondary Data

The neighborhoods under each *barangay*, their estimated population in 2020, the land area, and other relevant information were obtained from the involved *barangays* and the City Planning and Development Office (CPDO) of the City Government of Davao. Moreover, population per community and traced sewer lines for the year 2020 (when available) were obtained from each *barangay*. As the tracing of sewer lines by CPDO is still ongoing, data on all city sewer lines were not available.

In addition, clinical surveillance data for COVID-19 were constructed from the publicly available daily list of new cases released by the Department of Health, Davao Region [24]. The created database computed new and active cases (defined as new cases in the preceding two weeks) and cumulative incidence per 1000 persons.

Shapefiles from the CPDO and drainage network shapefiles from the City Engineer’s Office (CEO) of the City Government of Davao were obtained. Digital Elevation Models (DEM) of Davao from the Department of Science and Technology (DOST)-UP DREAM Phil-LiDAR Program were also used to generate the contour lines with a 1 m resolution, which was utilized as a base map. The sewer lines were filtered to only show the actual drainage networks in the city. However, they do not reflect the entirety of the sewersheds in Davao City. Therefore, the said sewer lines shown in the map are the current traced sewer lines as of 2020. The CPDO is still conducting feasibility studies for the sewer system and ongoing outlining the sewer lines. The team also collected the coordinates of the sampling sites using Garmin Montana 680, which were then imported into the base map as a comma-separated values (CSV) layer. All files were used in the open-source QGIS software version 3.16.11-Hannover for the sewershed map generation using the Geographic Coordinate System, World Geodetic System 1984 (EPSG: 4326).

### 2.3. Sample Collection and Processing

Wastewater samples were collected from the *barangay* sewer pipes in (1) 23-C, (2) 76-A Bucana, (3) Matina Crossing, and (4) Leon Garcia Sr., and from the creeks in (5) Mintal and (6) Tomas Monteverde between 8 AM to 3 PM once weekly for four weeks from 8 November to 12 December 2020. Weekly sampling was divided into two days, and *barangay* wastewater target sites in close proximity were sampled on the same day. For hydrologic analyses, qualitative descriptions of water flow, odor, type of waterway (sewer pipe, creek), and water body receiving the wastewater were noted. Mintal was chosen to represent a slow-to-moderately flowing site as the flow was never stagnant, whereas Matina Crossing was selected as a moderate-to-fast-flowing site. Samples for both SARS-CoV-2 detection and physico-chemical and microbial analysis were collected for each sampling date. In addition, discharge in these two sites was measured using a YF-S201 G1/2 water flow meter sensor with a range of 1–30 L/min from 23 November to 7 December 2020.

Two hundred and fifty milliliters (250 mL) of grab wastewater samples from each site were placed in sterile high-density polyethylene (HDPE) bottles, transported on ice to the College of Science and Mathematics, University of the Philippines Mindanao, and processed in a Biosafety Level 2 laboratory within four hours of collection.

The methods for processing wastewater samples were modified from Wu et al. [9]. Briefly, 40 mL of wastewater samples were sonicated at 20 kHz without pulse for two minutes. Next, samples were centrifuged at 4700× *g* for 30 min without pause to remove large particles. The supernatant was decanted and then passed through a 0.22 µm polyethersulfone (PES) syringe filter. Polyethylene glycol (PEG 8000) and NaCl were then added to the filtrate to a final concentration of 8% *w*/*v* PEG and 0.3 M NaCl. After vigorous agitation, the tubes were kept at 4 °C for 20 h and then centrifuged at 10,000× *g* for 90 min until a visible pellet formed. The viral pellet was then reconstituted with 100 µL DNA/RNA Shield (Zymo Research, Irvine, CA, USA) and stored at −20 °C until RNA extraction.

Physico-chemical parameters were obtained from all sampling sites by dipping the Hanna HI98194 multiparameter probes into a sample of wastewater collected in a flame-sterilized metal sampling cup. The readings were allowed to stabilize before recording the measurements for pH, practical salinity units (PSU), oxidation-reduction potential (ORP), total dissolved solids (TDS), and temperature. 

The IDEXX method detected and quantified total coliform and *Escherichia coli* (*E. coli*) from the water samples. Samples were collected using a flame-sterilized stainless-steel cup and washed three times to remove impurities. Approximately 100 mL of each water sample was collected in duplicates and transferred into sterilized glass bottles. After labeling, the bottles were then placed inside the cooler.

Microbiological analysis was performed in the laboratory immediately after the sampling. Aliquots of 0.1 mL and 1 mL were pipetted both for the total and *E. coli* coliform analysis. Each aliquot of 0.1 mL and 1 mL was placed in an IDEXX water sample vessel with 100 mL distilled water. The IDEXX™ Colilert™-18 reagent was added and diluted in the sample vessel. The mixture was transferred to the Quanti-Tray™ 2000 and sealed using the Quanti-tray sealer. The samples were incubated for 24 h at 35 ± 0.5 °C for total coliforms and *E. coli* in a dry incubator.

After incubation, the presence of total coliforms was determined by counting the yellow wells of the Quanti-tray that are equal to or greater than the comparator. The *E. coli* coliforms were determined by viewing the trays under ultraviolet light and counting the fluorescent wells equal to or greater than the comparator. The counts of the wells were referred to the MPN table to determine the number of coliforms present in the samples.

### 2.4. RNA Extraction and RT-PCR Analysis

Viral RNA was extracted from 100 μL resuspended viral pellets using the MagMax Viral/Pathogen Nucleic Acid Isolation Kit (Thermo Fisher Scientific Inc., Waltham, MA, USA) in a Kingfisher Duo Prime Purification System (Thermo Fisher Scientific Inc., USA). RT-PCR detection of SARS-CoV-2 was performed in duplicate using the Allplex^TM^ 2019-nCoV Assay (Seegene Inc., Seoul, Korea), which targets the E gene of Sarbecoviruses, the SARS-CoV-2-specific genes RdRP and N genes. Five microliters each of the 2019-nCoV MuDT Oligo Mix (MOM), RNase-free water, and 5X real-time one-step buffer, 2 μL of the real-time one-step enzyme, and 8 μL of RNA sample or controls (non-template control and positive control) were added for a total qPCR reaction of 25 uL. Thermal cycling was set at 50 °C for 20 min for reverse transcription, 95 °C for 15 min, and then 45 cycles of 94 °C for 15 s, and 58 °C for 30 s using the Applied Biosystems QuantStudio 5 real-time PCR machine (Thermo Fisher Scientific Inc., USA). Samples were considered POSITIVE for SARS-CoV-2 RNA when all three targets were amplified, or when either N or RdRP genes, or both N and RdRP genes were amplified with Ct below or equal to 40. These criteria were recommended by the manufacturers in the analysis and interpretation of the Allplex^TM^ 2019-nCoV Assay, and were also reported in other COVID-19 WBE studies [9,25]. However, if only the E gene was amplified, the sample was considered NEGATIVE for SARS-CoV-2 RNA. Linear regression analysis using the IBM SPSS Statistics v.21.0 software (Armonk, NY, USA) was used to assess if the Ct of the E, N, and RdRP genes were correlated with the RNA concentration and the computed weekly attack rate.

### 2.5. Whole-Genome Sequencing

To effectively maximize existing WGS resources, nine wastewater samples that were (1) positive for all targets (RdRP, N, and E genes) and (2) have Ct values below or equal to 39 were chosen for WGS. These criteria were modified from Crits-Christoph et al. [16], where only wastewater samples positive for the N gene with Ct values below or equal to 35 were subjected to WGS. SARS-CoV-2 amplicon enrichment was performed using the NEBNext SARS-CoV-2 companion kit (New England BioLabs, Ipswich, MA, USA) using the IDT 2019-nCoV v3 panel. PCR products for each sample, including a negative control, were diluted to 50 µL using 9.5 µL of cDNA from each of the two separate primer pool reactions and 31 µL of nuclease-free water. A total of 5 µL PCR dilution per sample was subjected to downstream processing consisting of a DNA end-repair step, barcode ligation, adapter ligation, and library clean-up following the published ARTIC nCoV-2019 sequencing protocol (LoCost) v3 [26]. Sequencing with the MinION Mk1B device (Oxford Nanopore Technologies, Ltd., Oxford, UK) was allowed to run for approximately 24 h.

### 2.6. Bioinformatic Analysis

Bioinformatic analysis was performed following the nCoV-2019 novel coronavirus bioinformatics protocol of the ARTIC Network [27], with several modifications made according to the actual experimental conditions. MinKNOW software was used to generate raw sequencing read files in FAST5 file format. The generated FAST5 files were subsequently base called and converted to FASTQ files using the Guppy Basecaller software v.4.4.1 (Oxford Nanopore Technologies, Ltd., Oxford, UK) set to high-accuracy mode base calling. The base called reads were then demultiplexed into their corresponding barcodes with the guppy_barcoder software. All barcoded reads were subjected to read filtering through the ARTIC guppyplex tool, retaining reads that are 400 (approximate amplicon size) to 600 bases in length. The remaining reads were then subjected to the medaka pipeline (set to r9.4.1 as this is the version of the flow cell used) set to use the ARTIC Protocol V3 (400 bp) primer set, which generated the assembled consensus sequences and performed variant calling. All consensus sequences were submitted to the Nextclade Web v.1.12.0 (https://clades.nextrain.org (accessed on 26 May 2022)) online server bioinformatics tool for the identification of viral mutations (single nucleotide polymorphisms or SNPs) using the SARS-CoV-2 isolate (Wuhan-Hu-1/2019) complete genome (GenBank Accession ID: MN908947) as the reference sequence [28]. Subsequently, the position of each identified SNP with respect to the sample from which they were detected was validated using Tablet v.1.21.02.08 (The James Hutton Institute, Scotland, UK) [29]. As the genome assemblies in this study were incomplete and highly fragmented due to the presence of many N base calls and hence sequence gaps, the threshold for variant calling was performed with a total of 30X coverage [30] on a specific base across all assembled consensus sequences to ensure high-quality variant calls. Clade assignment results from the Nextclade analysis were not considered because of low genome coverage and the high possibility of chimeric consensus genomes from mixed environmental samples.

### 2.7. SNP Mapping and Tracking

SARS-CoV-2 SNPs detected in wastewater were mapped via GIS coordinates using an open-source QGIS software v.3.16.11-Hannover (QGIS Association, http://qgis.org (accessed on 26 May 2022)), with Davao City *Barangay* shapefile containing the boundaries of the city’s 182 *barangays* and 2020 Population Density on the *barangays* of the sampling sites. The shapefile and the *barangay* population data were provided by the City Planning and Development Office of the City Government of Davao. The coordinates for each detected SNP were loaded into QGIS as a comma-separated values (CSV) layer. Point clustering was performed to prevent cluttering for SNPs that were detected on the same sampling site but at different time periods (i.e., D614G and R203K/G204R). Maps utilized the World Geodetic System 1984 (EPSG: 4326) coordinate reference system. In addition, point maps for previously reported SNPs and for unreported SNPs were created.

Wastewater SNPs detected in this study were screened against patient sequences derived from clinical surveillance in the Philippines that were submitted to the GISAID database. Essentially, two groups of GISAID sequences were retrieved. The first group consisted of sequences that were submitted before our sampling period, for which only sequences from the National Capital Region of the Philippines were available. The second group were sequences obtained during the post-sampling periods of January to December 2021 and were focused on Region 11 (Davao Region), to which Davao City belongs. Once downloaded, the GISAID sequence files were concatenated with the wastewater sequence files in Linux Ubuntu v.20.04.3 operating system (Canonical Ltd., London, England). The resulting FASTA file was submitted to Nextclade for SNP tracking. The findings were validated through manual inspection of the GISAID entries for SNPs.

## 3. Results

### 3.1. Site Characteristics

The largest community by land area is Mintal (7.68 sq. km), and the smallest is 23-C (0.21 sq. km). However, the population density was highest in 23-C, with 94,111 persons/sq. km., and lowest in Mintal, with 1929 persons/sq. km. (Table 1). Community wastewaters included in this study converged in community sewer lines that drained either into (1) Talomo River for the western Davao areas, such as wastewaters from Mintal; (2) Matina River, such as Matina Crossing; (3) Davao River, where we capture a subset from our collection in 76-A Bucana; or to (4) the Davao Gulf (Figure 1). 

The current drainage networks provided by CEO show the presently mapped and traced sewer lines in the city. This information has provided an estimate of the combined sewer water from residential wastewater, business wastewater, and street storm drains in the nearby areas of the site. An assumption is made that only a specific area of combined sewer water drains to the sampling sites and that a mixture of both untreated and inadequately treated wastewater from septic systems is sampled. This assumption can be backed by the contour line data, which assisted the research team in understanding the elevation of the city, the flow of the wastewater due to gravity, and the available drainage networks. The lack of sewer network data is evident for Mintal, whereas sparse sewer lines have been seen in Matina Crossing (Figure 1). 

In the six communities, 7 to 104 neighborhoods support 176 to 24,165 households. However, no shapefiles showing neighborhood boundaries in each community were available from the CPDO. Hence, the sewer networks running through individual neighborhoods could not be pinpointed clearly. In addition, COVID-19 cases were also reported at the community level and not by neighborhood; hence, the number of infected individuals per neighborhood and the contributing populations to the sewershed where wastewater samples were collected cannot be estimated. 

For the slow-to-moderately flowing Mintal site, the discharge was estimated at 1 to 4 L per minute or LPM. In contrast, the moderate-to-fast-flowing Matina Crossing yields discharge of about 16 to 28 LPM. The qualitative hydrologic and environmental site descriptions are detailed in Appendix A.

### 3.2. Physico-Chemical Parameters of Wastewater Samples

Physico-chemical parameters of the wastewater samples are summarized in Figure 2, Appendix A. During the time of collection, the pH of the sampling sites had a narrow range close to neutral, from 6.63 to 7.33. The oxidation-reduction potential (ORP) was mainly within the reduced potential, except for at Bucana where the ORP was oxidizing on more than one occasion. Monteverde yielded the most negative ORP values. Salinities were mostly freshwater, fluctuating narrowly within half of a unit for Bucana, Matina Crossing, and Mintal. A shift in salinity, by as much as nine PSU, at site 23-C for the November 13 sample collected at 1413H was distinct. Surface water temperatures ranged from 26.58 °C to 31.74 °C with minimal temperature variation in Mintal and Bucana and the highest variation at site 23-C. The Total Dissolved Solids (TDS) had a wide range from 102 mg/L to 8415 mg/L with the largest range at site 23-C.

Total coliform levels range from 534,000 to >2,419,000 MPN/100 mL with the lowest estimates in Mintal for each of the four weeks of sample collection (Figure 3 and Appendix A). Values were consistently high, even though the samples were diluted to 1:100 for Matina Crossing and Leon Garcia. *E. coli* estimates range from 114,650 to >2,419,000, with the lowest estimates in Mintal. Similar to Total Coliform, values were consistently high for Matina Crossing and Leon Garcia. 

### 3.3. SARS-CoV-2 Detection in Wastewater

Twenty-two (22) out of the 24 (91.7%) samples obtained from the six communities over the four-week course of the study tested positive for SARS-CoV-2, with RNA concentrations ranging from 7.8 to 40.2 ng/uL (Appendix A) and Ct values ranging from 29.41 to 39.73 (Table 2). Moreover, the detection rate for the individual genes was highest for RdRP at 87.5%, followed by N at 83.3%, and E at 62.5%. Fifteen samples (62.5%) were positive for all three genes. Two samples in this study were positive only for the RdRP gene, and one sample was positive only for the N gene. The E gene was consistently not detected in all samples from Mintal. However, no SARS-CoV-2 genes were detected in 23-C and Mintal samples, with low recovered RNA levels of 7.8 ng/uL RNA and 10.6 ng/uL RNA, respectively, during the week of 22–28 November 2020. 

### 3.4. Epidemiologic Assessment of SARS-CoV-2 Wastewater Surveillance

Except for Leon Garcia, all communities were at a high-risk level of COVID-19 transmission at least once during the study period (Figure 4). In addition, 23-C, Leon Garcia, Mintal, and Monteverde were categorized as low-risk communities during one of the four weeks of wastewater testing (Figure 4a,c,e,f). Therefore, none of the communities included in this study were categorized as CRITICAL or VERY HIGH risk for COVID-19 transmission during the entire wastewater sample collection. Moreover, no risk assessment was declared for the week of 15–21 November 2020.

Clinical surveillance data were aggregated per community during wastewater sample collection. Four hundred fifty new and 599 active cases were reported in the six communities during the wastewater testing period (Table 1). These cases cover only 14% of the new COVID-19 cases and 14% of the active COVID-19 cases reported in the entire Davao City for the same period. Over 44% of the new cases in the six communities occurred in 76-A Bucana. In addition, the weekly number of new and active cases was consistently higher for 76-A Bucana and Matina Crossing relative to the other communities (Table 1 and Figure 4). The weekly attack rates ranged from 5–127 new cases per 100,000 persons, with the highest attack rate recorded in Tomas Monteverde from 22–28 November (Figure 4).

SARS-CoV-2 RNA was detected in all sites in at least three out of four sampling points between November and December 2020, regardless of the Barangay Risk Category and even for barangays with a weekly attack rate as low as one new case per 10,000 population (Figure 4). Notably, SARS-CoV-2 RNA was detected in 23-C and Monteverde for the week of 5–12 December 2020, even when no new cases were reported for the corresponding week (Figure 4a,f). Conversely, SARS-CoV-2 RNA was not detected even if new cases were reported in 23-C and Mintal for 22–28 November 2020, with a weekly attack rate of 47 per 100,000 persons and 22 per 100,000, respectively (Figure 4a,e).

The Ct value for the E and N genes was negatively correlated with RNA concentration and attack rate (*p*-values < 0.05). In addition, the Ct value for the RdRP gene was negatively associated with the changes in the attack rate only (*p*-value = 0.04, Appendix A).

### 3.5. SARS-CoV-2 Sequencing in Wastewater

Nine wastewater samples from six sites qualified for whole-genome sequencing, with wide genome coverage. One sample from Matina Crossing (taken 9 November) had ~51.73% genome coverage. All other samples had a range of 1.14–24.53% genome coverage, indicating incomplete and highly fragmented assembled genomes (Appendix A). The spike gene coverage was zero for one sample (Matina Crossing, 30 November), and ranged from 9.34 to 53.98% for the rest of the samples. Non-synonymous SNPs were detected in only seven samples representing four communities and covering the four weeks of sample collection (Appendix A). 

Nextclade analysis detected a total of 29 non-synonymous mutations among all assembled consensus sequences, two of which were indels (Appendix A). Eleven of these mutations have been previously reported in patients, and a putative function has already been described for five of these. Among the known mutations in the spike gene for SARS-CoV-2 Variants of Concern (VOCs) and the Philippine P3 variant, only D614G was detected. Another SNP in the N protein, T247I, has been previously reported in cats alone [31]. On the other hand, 17 previously unreported non-synonymous SNPs were detected in the coding regions of ORF1a, ORF1b, ORF6, ORF7a, ORF8, and N.

Most of the mutations (14 of 29 SNPs) were only detected in one sampling site. In comparison, other SNPs were detected in multiple barangays (6–9 SNPs) (Figure 5 and Appendix A), the bulk of which was detected in the two most densely populated communities in the study, Leon Garcia and Tomas Monteverde (Figure 3). Eight SNPs affecting the N gene were detected in Monteverde in the first three weeks of the study (9 November to 4 December 2020) (Appendix A). In Leon Garcia, 4 SNPs affecting the ORF7a gene were detected in the last week of sample collection, 7–11 December 2020. D614G was found in all seven samples in all 4 weeks of the study.

Scanning the GISAID database showed that the mutations D614G of the S protein and P13L and R203K/G204R of the N protein have been detected among patients in the National Capital Region of the Philippines between March and July 2020 and in the Davao Region between January and August 2021, before and after the sampling period of this study (Appendix A). In addition, two SNPs, V21I of ORF7b and P383L of N, were clinically detected in the Davao Region during the post-sampling periods January to April 2021. The remaining SNPs have not been observed among GISAID sequences from the Davao Region.

## 4. Discussion

Wastewater-based surveillance of SARS-CoV-2 has been proven to be a useful surveillance tool for COVID-19 [32]. In this study, SARS-CoV-2 was detected in selected communities’ wastewater from Davao City, Philippines, regardless of the COVID-19 transmission risk category of the communities or presence or absence of reported cases. 

In some cases, SARS-CoV-2 RNA was detected in community wastewaters in Davao City even when no new cases of COVID-19 were reported for the corresponding week, which is in agreement with the results reported by Wannigama et al. [33] in Thailand. In this study, weekly attack rates in the involved communities from October to December 2020 are inversely correlated with SARS-CoV-2 Ct values. However, in some cases, SARS-CoV-2 RNA was detected even when the attack rate in the community was as low as one new case per 10,000 persons or even when no new cases were reported for the corresponding week. All active cases of COVID-19 in Davao City are immediately taken to one of the eleven Temporary Treatment and Monitoring Facilities (TTMFs) shortly after testing positive via RT-PCR [34], and over 70% of confirmed COVID-19 active cases in Davao City were asymptomatic by the end of 2020 [24]. Hence, SARS-CoV-2 RNA detected in community wastewaters may have come from pre-symptomatic or asymptomatic cases, or symptomatic individuals [33,35] who did not self-report to their local health monitoring unit. This finding can also be attributed to the differences in testing coverage in the communities, with more people tested in CRITICAL to HIGH-RISK communities [36]. 

Davao City has aggressive active-case finding, intensified contact tracing, and free community testing via RT-PCR, whereby even asymptomatic close contacts of confirmed COVID-19 cases are tested [36]. However, in cities or municipalities where clinical testing capacity and healthcare resources are low, both symptomatic and asymptomatic COVID-19 infections may be missed by clinical diagnostics (RT-PCR). In addition, even without testing resource limitations, COVID-19 infections in patients who purposely do not report having symptoms and do not get tested for COVID-19 because of fear of social stigma [37] or isolation and quarantine [10] will not be captured by clinical surveillance and contact tracing. Our pilot study on wastewater-based epidemiology in the Philippines offers a complementary public health surveillance method that can provide community infection data faster using fewer resources than individual RT-PCR testing [10,32] and without the social stigma. With long-term periodic community wastewater testing for SARS-CoV-2, WBE may present an unbiased snapshot of COVID-19 cases at the community level, whether asymptomatic, pre-symptomatic, or symptomatic [9]. Moreover, SARS-CoV-2 levels in wastewater can be used as an early warning marker for COVID-19 transmission trends, thus helping and directing health agencies where control programs should be intensified [7,11,12,13,35].

Here we also present the limitations of the wastewater detection method that was explored in this pilot study. The concentration of viral RNA in wastewater is a crucial step in WBE [37]. RNA recovery depends on several factors such as the volume of wastewater samples collected, sample processing (including viral concentration), and the RNA extraction method chosen [38]. In this study, 40 mL of wastewater was concentrated following a combination of PES filtration and PEG-NaCl precipitation methods, which were modified from Wu et al. [9]. For areas with low COVID-19 incidence, higher volumes of wastewater (e.g., 500 mL to 1 L) may increase the chances of SARS-CoV-2 detection [37]. Ahmed et al. [39] reported that using PEG precipitation could yield RNA recoveries ranging from 26.7 to 65.7% using a SARS-CoV-2 surrogate virus (murine hepatitis virus). Although RNA recovery was not computed in this study, RNA levels of wastewater-concentrated viruses were inversely correlated with the SARS-CoV-2 Ct values, suggesting that higher RNA yields generally increase the probability of SARS-CoV-2 wastewater detection. 

Although the commercially available AllPlex^TM^ 2019-nCOV assay (Seegene Inc., Korea) was developed for clinical applications using respiratory samples, we demonstrated its applicability for SARS-CoV-2 detection in feces-contaminated environmental samples, as has been done in Nepal [17]. The standard markers used for SARS-CoV-2 detection in wastewater are the N1, N2, and N3 genes and E Sarbecovirus gene [40]. Several studies reported detection rates of 70–100% when using the N gene as target [12,25,39,41,42]. Our pilot investigation suggests using either the N or RdRp gene instead, as they had consistently higher rates of detection (more than 80%) relative to E (62.5%) in community wastewaters. Ct values in our study were also within the previously reported range of 27 to almost 40 in wastewater [9,17,39,43,44,45,46]. Wurtzer et al. [8] reported 100% detection of RdRP genes when used as a confirmatory test for SARS-CoV-2 RNA presence. La Rosa et al. [47] reported contradicting findings where RdRP genes were not amplified in samples positive for the ORF1ab gene segment, possibly due to the higher limit of detection required for the RdRP assay (between 316–500 genome copies per reaction). While both studies used the RdRP primers developed by Corman et al. [48] in WWTP wastewater influents, they differed in the concentration method (ultracentrifugation vs. PEG precipitation) and the nucleic acid extraction method used (silica column-based extraction vs. magnetic bead-based extraction). The commercial RT-qPCR kit used in this study did not disclose the RdRP primer data. Additionally, genome quantitation standards for RT-PCR were not available during this pilot study. Hence, quantification of viral genome copies per volume of wastewater and, consequently, the determination of analytical sensitivity and specificity have not been performed, a limitation of our pilot study. Therefore, subsequent WBE studies must adhere to the Environmental Microbiology Minimum Information (EMMI) for publication of Quantitative Real-time PCR Experiments that describes essential and desired qPCR information, such as RNA yield, contamination assessment, inhibition testing, LOD determination, and repeatability to ensure the validity and reliability of qPCR studies [49].

This pilot application identified a unique set of challenges in the implementation of wastewater surveillance in the Philippines. COVID-19 WBE has been reported in over 50 countries worldwide, 94% of which belong to the high- and upper-middle-income countries with centralized wastewater treatment plants (WWTPs) [7]. With WWTPs, accurate data about the population served by the specific treatment plant are available, and biological, physicochemical, and hydrologic data are regularly monitored, permitting near-real-time spatio-temporal trend analysis on COVID-19 transmission in the served population [37]. Unlike high- and upper-middle-income countries, most cities in the Philippines, including Davao City, do not have centralized wastewater treatment facilities for many residents. Over 90% of households in Davao City are connected to private or communal septic systems [21]. Another challenge in this pilot study was obtaining reliable estimates of the contributing population to the wastewaters collected since data on sewer pipe networks in the communities were incomplete as of writing. Cities without centralized WWTP that plan to implement WBE, whether for COVID-19 or other water-related infectious diseases, must ensure that a comprehensive map of sewer networks is first generated to accurately identify the population contributing to the wastewater. However, WBE can still be recommended for non-intrusive targeted site testing for COVID-19 infections even in areas without a centralized WWTP. For example, COVID-19 wastewater surveillance in residential buildings was performed in Singapore [14] and Hong Kong [50] as a screening test. Their respective health agencies implemented compulsory testing of individual residents and isolation of infected persons in buildings where wastewaters tested positive for SARS-CoV-2.

We are currently evaluating the contribution of physico-chemical, hydrologic, and anthropogenic factors to the SARS-CoV-2 RNA recovery from community wastewaters in Davao City. Each wastewater sample collection site has a distinct physico-chemical, hydrologic, and anthropogenic profile that may affect RNA preservation [22], and consequently, SARS-CoV-2 RNA detection. For example, the sample collection site in Mintal is a natural creek with consistently low fecal coliform load (less than 500,000 MPN/100 mL), with pH between 6.63 to 6.99 (lower compared to the other sites), low TDS (102 to 171 ppm), high ORP (6.63 to 159.10 mV), and a water temperature that ranged from 27.49 to 27.85 °C. The weekly sample collection for this site was between 1300 to 1500 H and the discharge was between 0.05 to 0.19 gallons per minute. Mintal is the least-densely populated community among the sampling sites. Mintal was categorized as a MODERATE or HIGH-risk community in three of the four weeks of the study, and SARS-CoV-2 RNA was consistently detected during these three weeks, even as the RNA concentration fell to as low as 10.2 ng/uL. In contrast, no SARS-CoV-2 genes were detected in Mintal from 22–28 November 2020, when Mintal was categorized as a LOW risk community, even when viral RNA concentration was similar to the other weeks (10.6 ng/µL). The absence of SARS-CoV-2 RNA in sites where new cases are reported may be caused by the varied physico-chemical [22,51], hydrologic, and anthropogenic factors present in a specific site [52,53]. Since the Mintal wastewater is more homogenous, it is speculated that viral RNA, albeit at low levels, was preserved. Thus, the combined effects of physico-chemical, hydrologic, and anthropogenic factors on viral RNA recovery from wastewater in individual sites must be investigated [22], especially in areas with no centralized WWTPs. Ct value cut-offs for RT-qPCR detection of SARS-CoV-2 in wastewater may have to be adjusted in consideration of these combined effects on RNA recovery.

In most cities, the sewage flow rate is most significant during the morning and the evening [37]. During sample collection, qualitative descriptions of flow were indicated and it was noted that Matina Crossing attained the fastest flow, whereas Monteverde generally showed slow to stagnant flow (Appendix A). A 24 h composite sampling can catch these peak flows with the most reliable mean viral levels in the wastewater [37]. However, composite samples may also result in reduced viral titers compared to grab sampling [12,37] because of the dilution of viruses [17]. In addition, composite sampling is also logistically challenging, especially without expensive automated wastewater collectors. In our pilot study, grab samples were collected between 8 AM and 3 PM, depending on the site’s distance from the laboratory and the changing of the tides (for sites near the Davao Gulf). For future studies, we suggest that the collection of grab samples during peak hours of human toilet use (e.g., 7 AM to 9 AM) as an alternative method that can improve the chances of detecting fecally shed SARS-CoV-2, especially in low-incidence and low-resource areas [17].

The discharge of the water also plays a role in the general physico-chemical character of the surface waters [52]. Monteverde was stagnant in the four occasions of sample collection for the four-week sampling campaign. The residence time of the wastewater in Monteverde is also expected to be longer, accumulating organic material from the sewershed over time. Some of these organic-rich materials are deposited at the bottom, where it is likely that suboxic to anoxic conditions prevail. This condition is evident in the intensely negative ORP readings even at the surface. Monteverde site was also distinctly identified as the site where a pungent smell of ammonia was observed. Nitrogen from organic matter in the form of fecal material, amino acids, or urea is hydrolyzed by chemical or bacterial-mediated processes to form ammonium (NH_4_+) in neutral to acidic pH or ammonia (NH_3_) in basic pH of the waters [54]. This coincides with its low ORP readings (the lowest among the sites) showing that it has the most reducing conditions. The fast-flowing wastewater in the Matina Crossing site has low residence times. Therefore, it is expected to represent the physico-chemical properties closer to the community’s combined wastewater, fecal load, and SARS-CoV-2 load at the time of sampling. Other sites with slower discharge will also have longer residence time and can represent longer times of pooled organic matter from the community. A snapshot sampling of these slower-moving waters will represent longer records of viral RNA loading compared to faster flowing waters with shorter water residence times [55]. For forecasting purposes, we recommend the Matina Crossing site for further WBE monitoring, since this is also one of the sites with high urban activity. We also recommend some moderate to slow-flowing waterways for further WBE monitoring.

Sample collection at sites close to the Davao Gulf (i.e., 76-A Bucana, Leon Garcia, Monteverde, and 23-C) was influenced by tidal changes. During high tide, the sewer pipes are partially submerged in seawater (or riverine water at 76-A Bucana), and the direction of flow is inward, which reverses during low tide. Besides dilution, saltwater intrusion into wastewater may hasten the decay of enveloped viruses like SARS-CoV-2 [56]. Hence, sample collection was carried out only during low tide to reduce wastewater dilution with seawater. Even with this effort, it is evident that there is still slight mixing of saltwater at site 23-C, evident from the increase in salinity to 10 PSU for the November 13 sample, which was collected 1417 H during a high tide that rose 1 m above the mean lower low water. With the influence of the high tide, the seawater was also exposed to hours of intense heat, which also explains the abnormal temperature increase of 4 °C, raising the water temperature to almost 32 °C. The temperature increase, which is likely in the minimum to moderate range of change, may add to the decay of the organic components in the water [57,58]. Entrainment of viral RNA from the densely populated shore communities to more inland sampling sites, such as 23-C and Bucana, during high tide will still underestimate yields because the exposure to environmental factors, such as salinity, degrading enveloped viruses such as SARS-CoV-2 [42]. For example, the November 13 sample from 23-C possibly provided minimum viral yield compared to a sample that may have been collected during low tide. Sample collection times should therefore consider tidal conditions, wherein high tide should be avoided.

Fecal loading of waterways, which estimates the contributing population, can influence the viral load of the water sample [22]. Anthropogenic factors such as industrial discharge and livestock waste have been shown to alter the water quality and may influence nutrient loads, coliforms, and chemical composition [59]. O’Brien and colleagues [60] also found that the abundance and diversity of viruses in wastewater varied with their environment. In this study, the characteristics of community wastewater varied across the sites. For instance, the Mintal site receives domestic wastewater, whereas Matina Crossing, 23-C, and Bucana, which are communities with denser populations and more economic activities, receive a mixture of domestic and industrial wastewater. The population density in the upper watershed, like the Mintal site, is relatively low compared to the lower watershed, such as Bucana, 23-C, Leon Garcia, and Matina Crossing. Total Coliform enumerated in the primarily domestic and agricultural areas in the upper watershed may include fecal material from other animals and fertilizers. The natural waterway provides opportunities for interaction with soils and streambeds that can either dilute or add to the fecal load or degrade the fecal load, resulting in relatively lower total coliform MPN [61]. The fecal load in the Mintal site is consistently the least among the sites sampled. The ratio of *E. coli* to total coliform, which ranges from 0.12 to 0.66, can be calculated since the sample dilution is sufficient to estimate the concentration. Although varied, this range of values is relatively low, suggesting a lower contribution of fecally derived coliforms in the Mintal site compared to the other sites. In the lower watershed, even when the level of *E. coli* values which are much less than total coliform still exceeds the values in the enumeration tables, most are reported as >2,419,000. These results indicate the release of untreated wastewater with high fecal load from humans and other warm-blooded animals [62]. Two other sites for week two yield concentrations with reasonable estimates for MPN counts. Their *E. coli* to total coliform ratios were calculated as 0.70 and 0.71 for Bucana and Monteverde, which are higher and narrower in range. More datasets need to be analyzed to know if there are trends in these ratios. Nevertheless, fecal coliform has the potential to estimate the contribution of human and/or animal waste in the wastewater at specific collection sites.

Genomic surveillance in wastewater was attempted in this pilot work. An average read depth of genome coverage was calculated to be ~20×, indicating that SARS-CoV-2 genomes were sequenced reasonably and accurately [63]. Even as the ARTIC 2019-nCov sequencing protocol v3 (LoCost) [27] method was used successfully in several COVID-19 WBE studies in WWTP [64,65,66], fragmented genomes were generated in our study. This may be due to (1) the problematic nature of the untreated community wastewater samples, which may have been composed of degraded viral genetic material [37], and (2) the high Ct values (above 30) of the samples, which are known to be technically challenging for whole-genome sequencing [16]. Nevertheless, we analyzed the sequence data as an exploratory frame of reference for future investigations. The threshold for variant calling was performed with a total of 30× coverage to ensure high-quality analysis despite the fragmented sequences [28]. According to Coil et al. [67] near-complete SARS-CoV-2 genomes from environmental swab samples can be generated by adding a concentration and PCR clean-up step before WGS and duplicating barcoding reactions. This modification must be explored in WGS using wastewater in future studies, considering that surface swabs and wastewater are different matrices with varying attributes that may affect overall WGS success.

SARS-CoV-2 SNPs from Davao City wastewaters were found in Philippine clinical cases that occurred before and after our sampling period. Some of these clinical SNPs were detected as early as the first three months of the country’s outbreak (March to June 2020) such as the globally or Asian-dominant ORF1a T2016K, spike D614G, and nucleoprotein P13L, R203K, and G204R [68,69,70,71], and the less frequently reported mutations ORF1a L3930F, ORF1b A88V, and ORF7b V21I [72]. Notably, our wastewater surveillance was able to detect SNPs P383L and V21I in November–December 2020, even before their first clinical detection a month later in the Davao Region (January 2021). This finding highlights the utility of WBE–WGS in early detection of recently introduced SARS-CoV-2 variants that are circulating in the population. In addition, other SNPs such as ORF1a A1432V, ORF1b P234L, and nucleoprotein S197L have been reported elsewhere in the world [73,74,75].

Despite the low genome coverage, we were able to detect mutations that were not yet reported by clinical surveillance (Appendix A). One of these is the V117I SNP at ORF8, an accessory protein for viral capsid formation that is also implicated in the host immune response towards SARS-CoV-2 [76,77], and the putative alternative start codon of ORF7a, a phenomenon reported for other viruses such as the Sindbis virus [78]. In addition, several SNPs were detected in the densely populated communities of Leon Garcia and Monteverde. Considering the high mobility of people and increased transmission of the virus in dense populations [79,80,81], the introduction or emergence of new variants is more likely in these areas. The three SNPs at the N gene (R203K, G204R, S197L) have been associated with increased transmission and virulence, increased virus fitness [71], and increased antigenicity with possible consequences in antibody production [70,82]. Finally, nucleoprotein T247I, detected in Monteverde, is of interest because it has been associated with cat isolates alone [31], implicating either animal carriers of SARS-CoV-2 in the community or human variants that carry animal-associated SNPs. Case reports from Madrid, Spain [83] and Switzerland [84] reported SARS-CoV-2 in pets (cats and dogs) in households with COVID-19. These findings show that WBE WGS can obtain genomic information about disease dynamics and community transmission in a rapid and resource-efficient manner [17].

Altogether, our pilot genomic surveillance in wastewater confirmed the presence of previously reported mutations while identifying mutations not yet registered in clinical surveillance. Wastewater genomic surveillance (WGS) can therefore facilitate early detection of mutations using fewer resources, given that pools of individuals are represented in every wastewater sample. However, the current strategy has large room for improvement, considering the fragmented nature of the sequence data. Nevertheless, WGS has the potential to complement clinical surveillance especially in the Philippines, where only representative clinical samples are included in the national genomic biosurveillance program [85], thereby introducing the possibility of identifying other relevant circulating SARS-CoV-2 variants.

This first report on COVID-19 wastewater surveillance in the Philippines and the Western Pacific region provides insights and recommendations that may be useful to countries or regions with similar experiences. 

## 5. Conclusions

SARS-CoV-2 RNA was successfully detected in community-level wastewater from the six barangays of Davao City, wherein wastewater-based epidemiology (WBE) has the potential to capture asymptomatic or unreported cases in the community. This will play an important role in community monitoring of COVID-19 as we slowly transition to the post-pandemic stage. However, the lack of a sewershed map in the sampled areas in Davao City is a major challenge which needs to be integrated in the planning for WBE, such as providing assistance to the government in the generation of maps. Nevertheless, a combined analysis of wastewater physico-chemical parameters and anthropogenic conditions provided critical insights on the implementation of WBE in the investigated areas. We also detected previously reported and newly identified SNPs covering several SARS-CoV-2 genes, despite the fragmented genomes recovered from wastewater. These findings provide a baseline framework for the application of WBE in low-sanitation metropolitan areas such as Davao City, Philippines, even for other infectious diseases of public health relevance. We recommend that a similar multifaceted assessment be conducted in other low-sanitation areas prior to WBE implementation.

## Figures and Tables

**Figure 1 ijerph-19-08789-f001:**
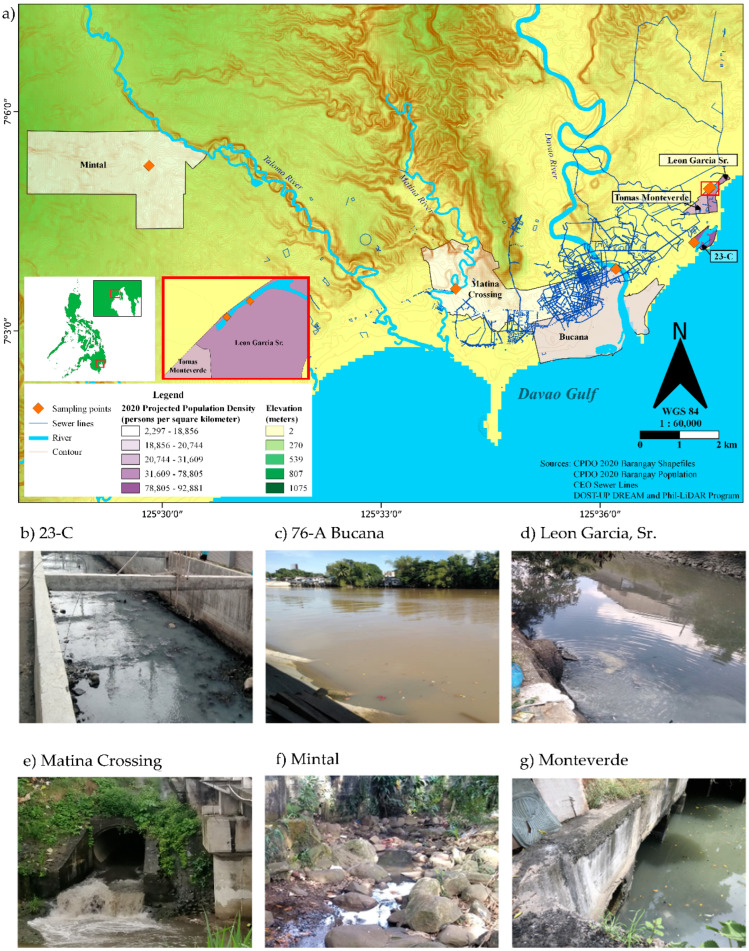
Wastewater outfall and sampling sites in Davao City. (**a**) Location of the six *barangays* and sampling sites with respect to the major draining natural bodies of water. Sampling site conditions showing the access points to sewer pipes in (**b**) 23-C, (**c**) 76-A Bucana, (**d**) Leon Garcia, (**e**) Matina Crossing, (**f**) Mintal, and (**g**) Monteverde. The sampling *barangays* were projected on the map using the shapefiles and 2020 Population Density data (persons/sq. km.) from the City Planning and Development Office (CPDO) of the City Government of Davao. Sewer lines (blue line networks on the map) were also obtained from the City Engineer’s Office (CEO) of the City Government of Davao. The contour lines (meters; shown in thin brown thin lines on the map) were obtained from the Department of Science and Technology, University of the Philippines Disaster Risk Exposure Assessment for Mitigation and Phil-LiDAR Program.

**Figure 2 ijerph-19-08789-f002:**
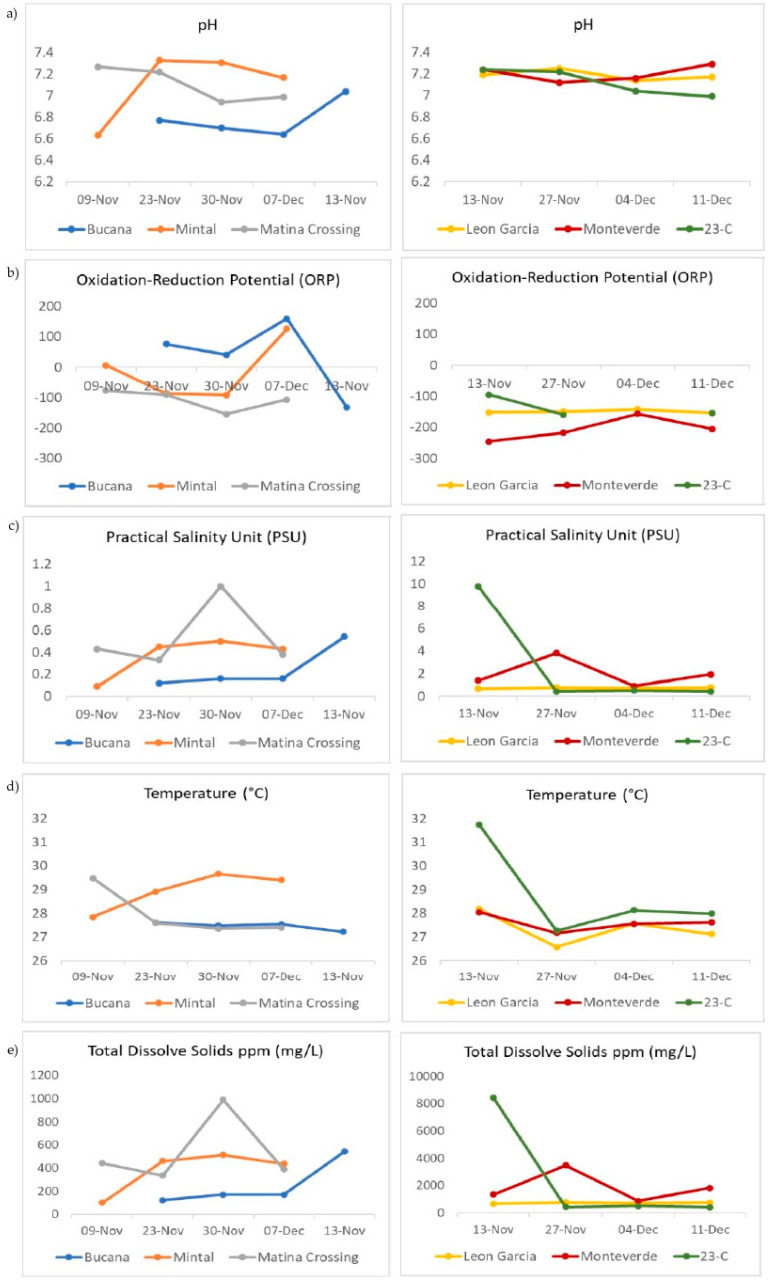
Physico-chemical parameters measured using Hanna multiprobe (hi-98194) during sample collection. The ranges of the specific parameters measured and the accuracy of measurements are indicated in the following: (**a**) pH is 0.00 to 14.00 pH with a resolution of 0.01 pH, accuracy of ±0.02 pH, (**b**) Oxidation-Reduction Potential (ORP) values are ±2000.0 mV with a resolution of 0.1 mV, accuracy of ±1.0 mV, (**c**) salinities within 0.00 to 70.00 PSU with a resolution of 0.01 PSU, accuracy of ±2% of reading or ±0.01 PSU whichever is greater, (**d**) temperature is from −5.00 to 55.00 °C with a resolution of 0.01 °C, accuracy of ±0.15 °C, and (**e**) the Total Dissolved Solids (TDS) is 0 to 400,000 ppm (mg/L), with the maximum value depending on the TDS factor, which was set automatic ppt (g/L), accuracy of ±1% of reading or ±1 ppm (mg/L) whichever is greater.

**Figure 3 ijerph-19-08789-f003:**
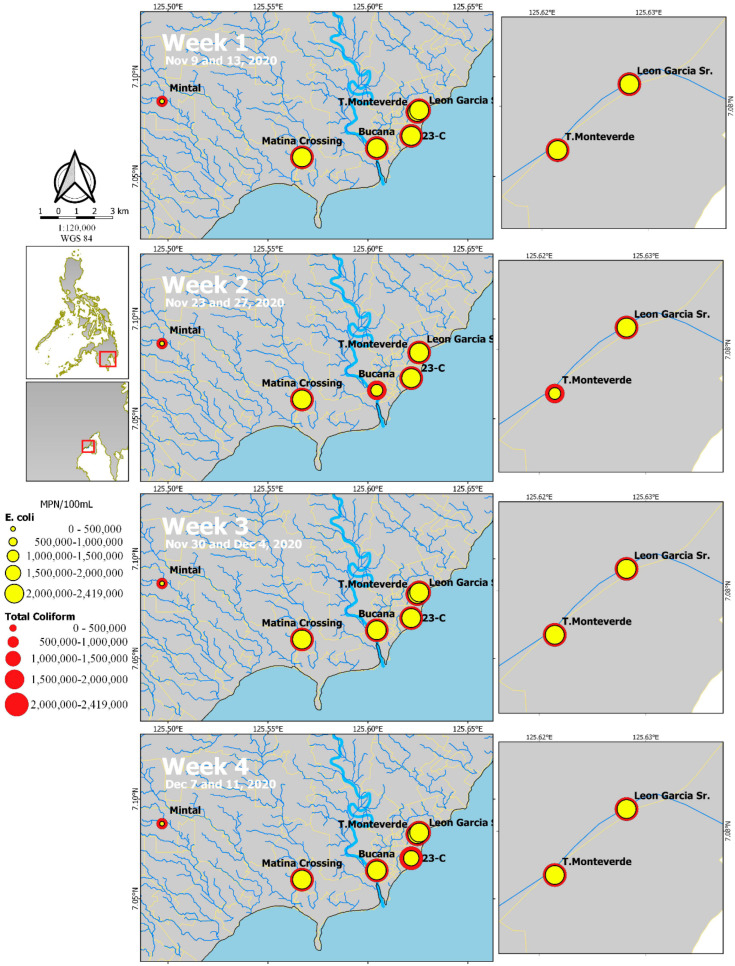
Total coliform and *Escherichia coli* (*E. coli*) quantification in Most Probable Number (MPN) using the IDEXX method for the sampling sites indicated.

**Figure 4 ijerph-19-08789-f004:**
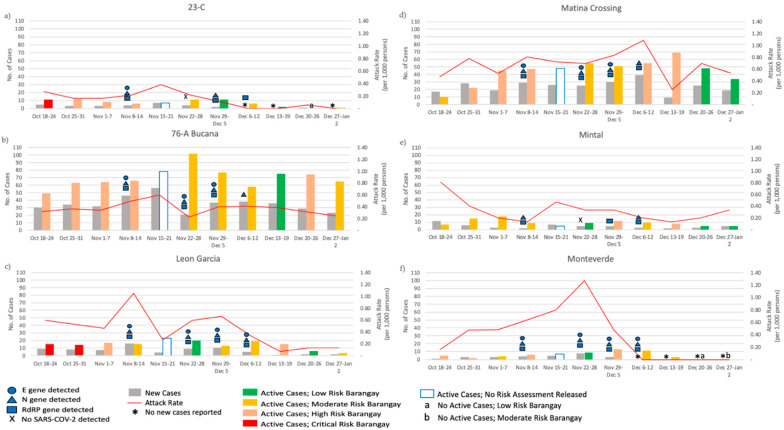
Detection of SARS-CoV-2 in wastewater and COVID-19 clinical surveillance status and in six communities of Davao City, Philippines from 18 October 2020 to 2 January 2021. SARS-CoV-2 RNA was detected in community wastewater from (**a**) 23-C, (**b**) 76-A Bucana, (**c**) Leon Garcia, (**d**) Matina Crossing, (**e**) Mintal, and (**f**) Monteverde in at least three of the four weeks of testing.

**Figure 5 ijerph-19-08789-f005:**
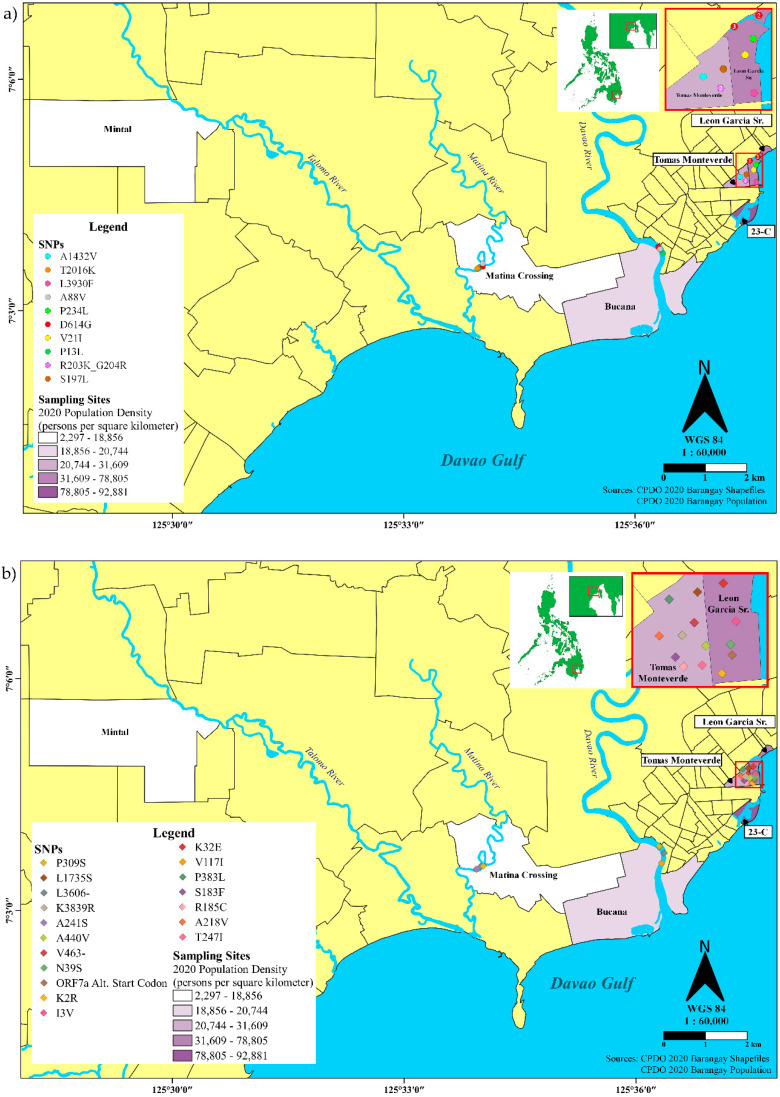
Geographic distribution of SARS-CoV-2 SNPs detected in wastewaters of Davao City, Philippines on November-December 2020 that have been (**a**) previously reported, or (**b**) unreported from clinical cases in GISAID or literature.

**Table 1 ijerph-19-08789-t001:** Community and sampling site characteristics.

Community	2020 Est. Population	Land Area (sq. km.)	Population Density (Persons/sq. km.)	Source of Wastewater	Water Body Receiving Wastewater	New COVID-19 Cases ^1^	Active COVID-19 Cases ^1^
23-C	18,474	0.20	94,111	Sewer pipe	Davao Gulf	17	23
76-A Bucana	94,074	4.02	23,409	Sewer pipe	Davao River	198	264
Leon Garcia	15,296	0.22	68,224	Sewer pipe	Davao Gulf	44	59
Matina Crossing	36,342	5.29	6866	Sewer pipe	Matina River	149	196
Mintal	14,820	7.68	1929	Natural creek	Talomo River	22	31
Monteverde	6404	0.21	31,042	Natural creek reinforced with boulders	Davao Gulf	20	26
Entire Davao City	1,816,987	2440.00	744.67	-	-	3120	4379

^1^ Only for the period 8 November to 12 December 2020.

**Table 2 ijerph-19-08789-t002:** SARS-CoV-2 detection in six communities of Davao City, Philippines from 9 November–12 December 2020.

Gene Marker	Communities ^1^	
	Positive (%)	Ct Range ^2^
E	15/24 (62.5%)	29.95–39.73
N	20/24 (83.3%)	29.41–38.74
RdRP	21/24 (87.5%)	31.26–38.89
Overall	22/24 (91.7%)	29.41–39.73

^1^ Total of 24 samples. ^2^ Ct values of positive samples.

## Data Availability

Not applicable.

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
