# Peer review of "Multifaceted Assessment of Wastewater-Based Epidemiology for SARS-CoV-2 in Selected Urban Communities in Davao City, Philippines: A Pilot Study"

_ijerph, 2022, doi:10.3390/ijerph19148789_

Round 1

Reviewer 1 Report

The current manuscript reports “Wastewater-based epidemiology and whole-genome sequencing for community-level surveillance of SARS-CoV-2 in selected urban communities in Davao City, Philippines: A Pilot Study”. The manuscript although written technically, it lacks any novelty.

Major concerns:

  1. Technically there is nothing new to add to the already existing reports with regards to wastewater-based epidemiology (WBE). Mere analysis of WBE in low-sanitation countries can not provide new insights except the fact that is first report from Phillipines.
  2. Line 155-156: As per kit instructions if only E gene gets amplified then sample has to be considered presumptive positive. However, authors have considered those samples as negative for SARS-CoV-2 RNA in the text but positive in Table S1. There is conflict in what author has written in text and depicted in Table S1.
  3. Previous studies (e.g. DA Coil et al., 2021) have already shown in their reports that viral sequences could be retrieved from environmental samples that were negative by qRT-PCR. Why authors did not consider all samples for sequencing.
  4. The way data has been represented and reported it is not quite clear as to whether the authors made sure that the samples were tested in replicates?
  5. The authors have not made any attempt to establish a correlation between expected number of infected individuals based on experimental data and actual no. of infected cases reported.
  6. The manuscript also fails to highlight what and which mutations are getting detected more and more with time and zone wise locations of such spread with time.
  7. Authors should try to explain SARS-CoV-2 RNA detection in community wastewaters even with no reported new cases and absence of SARS-CoV-2 RNA when cases are being detected in the corresponding barangay.
  8. The authors must explain the selection criteria applied to all the samples which were considered for WGS.
  9. The authors should have done a comprehensive discussion with regards to the differences in detection rates of all three genes of COVID-19.
  10. Why did the authors fail to include or mention properly the RNA recovery in the study although author mentioned it for other studies?
  11. The authors fail to highlight what relevant information is added to the existing knowledge? How this study is an advancement in the scientific knowledge in the domain.
  12. Line 284-287: indicates the information about the Genome coverage. This information needs to be included in a separate table with % coverage of the Spike gene.
  13. As the author has mentioned that only one sample has achieved a genome coverage of ~51.73 %, whereas all other samples have a range of 1.14%-24.53%, which is too much low to predict the variants. Sequencing needs to be done again to achieve higher coverage.
  14. Only a single spike mutation is detected from data i.e D614G, which is a common mutation among all the variants of SARS-CoV2. There is no novelty in terms of sequencing results. How sequencing results are different from Rt-PCR if no unique mutations were detected from the data.!?
  15. Why GISAID data has been divided into two groups!! where is the correlation? The author has only documented the no. of sequences by mutation.
  16. My major concern is why authors have done NextClade analysis for the metagenomic samples! Since ARTIC bio info pipeline is designed for only clinical single isolate sample., Assembling a genome would be a wrong idea until the assembler resolves the assembly at the strain level. which is quite difficult for the current assemblers also it requires higher genome coverage.
  17. What is the depth coverage of mutations !!

Minor concern

  1. barangay is written sometimes in italics, sometimes in normal font, authors should follow one font throughout manuscript.
  2. Line 76-77: provide reference for Paris study.
  3. Line 146: What is MOM?
  4. Line 162,163: Add – between CoV and 2.
  5. Line 180: Thru, Informal writing!!
  6. Line 259: remove space beforeCoV.
  7. Line 296-297:few is a ambiguous word used in text, clearly mention numbers.
  8. Line 284: Sequencing details are missing. (Check Table S5).
  9. Line 341: provide space after reference number.
  10. Line 349: Enter reference before full stop.
  11. Line 414: Remove full stop after reference.
  12. Figure 2, e: Last day of January is missing at x axis. In legends if both a and b depicts same category, then why it is named twice.
  13. Table 2: samplings should be replaced with samples, remove on from table heading.
  14. Table 3: specific date should be mentioned in heading, what is the difference between not yet reported and not yet investigated.

Author Response

Dear Reviewer 1,

Kindly see the attached file for the summary of revisions. Thank you.

Reviewer 2 Report

The work described wastewater based epidemiology of SARS-CoV2 by using whole genome sequencing. The authors found SNP clinically described and new SNP in the Philippines. They also detect SARS-CoV2 RNA in communities with no reported new cases. Results are clear and understandable, and illustrate the possibility to use wastewater sequencing based epidemiology to monitor spread of COVID19. Several teams have yet worked on this topic but it is the first one carried out in Philippines. This works deserves to be published after minor revisions.

1- What are the criteria to generate the consensus sequence (coverag and identity thresholds)?

2- LINE 220: persons /sqm--> is the unit right?

3- FIGURE 2: What do the stars in fig2e and fig2f mean? 

4-The authors have detected new SNP, never described in Philippines, It would be valuable for  to discuss the perspectives and the interest of this detection. I suggest the same for the SNP described in cats.    

Author Response

Dear Reviewer 2,

Kindly see the attached file for the summary of revisions. Thank you.

Round 2

Reviewer 1 Report

Despite offering revision, authors are unable to justify the comments raised by me, so I feel that there is a lot of improvement required in the sequencing and analysis part to grant publication of this study in this journal. Hence I would suggest rejecting the paper for further thorough improvements. Its been almost more than 2.6 years of the covid pandemic, all the research protocols and methods are well tested and validated, therefore improper research results and methods should not be acceptable.

Author Response

Dear Reviewer 1,

Our team reevaluated the manuscript considering your feedback. After deliberation, we decided to repackage our paper to highlight the relevance and novelty of the work. We have refocused the paper on the following major highlights: 

  1. the pilot implementation of SARS-CoV-2 WBE despite the methodological and logistical

    challenges in low-sanitation areas; and

  1. the possible contributions of physico-chemical parameters and anthropogenic conditions on

    the design of WBE in the selected areas and on RNA recovery.

We also reported our exploratory attempt on wastewater WGS to demonstrate its potential application as a complementary surveillance tool, considering that genomic surveillance in the Philippines is limited in scope.

Hence, the title was changed to “Multifaceted Assessment of Wastewater-Based Epidemiology for SARS-CoV-2 in Selected Urban Communities in Davao City, Philippines:  A Pilot Study.”

The results and discussion parts were reorganized considerably to improve coherence of ideas and highlight the importance of the findings.

We firmly believe that our research output would expose the challenges in implementing WBE in the Philippines and even in other countries/sites that do not have centralized wastewater treatment plants (WWTPs).

Hoping for your kind consideration. Thank you very much.